

# Impact of seasonal flooding and hydrological connectivity loss on microbial community dynamics in mangrove sediments of the southern Gulf of Mexico

Mirna Vázquez-Rosas-Landa[1,*], Rosela Pérez-Ceballos[2,3], Arturo Zaldívar-Jiménez[4], Stephanie Hereira[5], Leonardo Pérez González[1], Alejandra Prieto-Davó[6], Omar Celis-Hernández[2,3] and Julio Cesar Canales-Delgadillo[2,3,*]

[1] Unidad Académica de Ecología y Biodiversidad Acuática, Instituto de Ciencias del Mar y Limnología, Universidad Nacional Autónoma de México, Mexico, Mexico
[2] Instituto de Ciencias del Mar y Limnología Estación El Carmen, Universidad Nacional Autónoma de México, Ciudad del Carmen, Campeche, Mexico
[3] Secretaría de Ciencias, Humanidades, Tecnologíae Innovación (SECIHTI), Mexico, Mexico
[4] Asesoría Técnica y Estudios Costero, Merida, Yucatan, Mexico
[5] Laboratorio de Interacciones Bióticas, Centro de Investigación en Ciencias Biológicas, Universidad Autónoma de Tlaxcala, Tlaxcala, Mexico
[6] Unidad de Química en Sisal, Facultad de Química, Universidad Nacional Autónoma de México, Puerto Abrigo, Yucatan, Mexico
[*] These authors contributed equally to this work.

Corresponding author
Julio Cesar Canales-Delgadillo,
jccanalesde@conahcyt.mx

## ABSTRACT

**Background**. Mangrove ecosystems play essential roles in coastal resilience, carbon sequestration, and biodiversity but are under increasing threat from anthropogenic pressures. This study explores the impact of hydrological variability on microbial communities in mangrove sediments of the southern Gulf of Mexico.

**Methods**. We employed 16S rRNA sequencing to assess microbial diversity and function across different hydrological zones, seasons, and sediment depths at Estero Pargo.

**Results**. Our results show that microbial community composition is significantly influenced by hydrological conditions, with distinct microbial assemblages observed across the fringe, basin, and impaired zones. Seasonal variations were particularly pronounced, with higher microbial diversity during the flood season compared to the dry season. Depth also played a critical role, with surface layers (5 cm) predominantly featuring aerobic microbial communities, while deeper layers (20–40 cm) harbored anaerobic taxa such as *Bathyarchaeota* and *Thermococcaceae*. Notably, the impaired zone showed enrichment in genes related to denitrification and sulfur oxidation pathways, indicating strong microbial adaptation to reduced environments. These findings highlight the intricate interactions between microbial dynamics and environmental factors in mangrove ecosystems. Understanding these relationships is crucial for developing effective conservation and management strategies that enhance mangrove resilience in the face of global environmental changes.

## INTRODUCTION

Mangrove ecosystems, critical components of tropical and subtropical coastal landscapes, play essential roles in carbon sequestration and influence global weather patterns (*Lee et al., 2014*; *Chatting et al., 2022*; *Song et al., 2023*). Despite their ecological importance, these ecosystems are vulnerable to threats from human-induced alterations such as altered river flows and extensive deforestation (*Akram et al., 2023*). Microorganisms within these ecosystems are key, facilitating nutrient cycling, disease resistance, and stress tolerance, thereby sustaining ecosystem health and functioning (*Palit et al., 2022*; *Holguin, Vazquez & Bashan, 2001*; *Akram et al., 2023*; *Bashan & Holguin, 2002*; *Vovides et al., 2011*; *McKee & Faulkner, 2000*; *Woodroffe et al., 2016*). However, the full extent of their role, especially in ecosystem restoration, is not yet fully understood.

In the rhizosphere and root tissues of mangroves, microorganisms perform critical functions such as nitrogen fixation, enhancing nutrient availability in typically nutrient-poor coastal environments (*Purahong et al., 2019*; *Sjöling et al., 2005*; *Vovides et al., 2011*). These interactions not only promote mangrove growth but also enhance resilience to environmental stressors like salinity and phytotoxin (*Lai et al., 2022*). Furthermore, diverse bacterial genera such as *Agrobacterium*, *Bacillus*, and *Pseudomonas* offer protective effects against pathogens, highlighting the potential for microbial applications in mangrove conservation (*Bonaterra et al., 2022*).

Tidal fluctuations in mangrove ecosystems dictate crucial factors such as nutrients, oxygen levels, dispersal mechanisms, and salinity. These elements are essential for the well-being and distribution of mangrove species and their associated microbial community. Human-induced alterations to hydrological patterns can disrupt these ecosystems, leading to reduced tidal flushing, elevated salinity, sedimentation, and nutrient imbalances (*Kamali & Hashim, 2011*; *Pérez-Ceballos et al., 2020*). Hydrological dynamics also shape the distribution and composition of microbial communities in mangrove habitats, with tidal inundation introducing diverse microbial populations from neighboring areas, enriching soil microbiota. Changes in hydrological patterns can disrupt these microbial communities, affecting nutrient availability, decomposition rates, and ultimately mangrove health (*Mai et al., 2021*; *Thomson et al., 2022*).

Acknowledging the holobiont concept, where mangrove trees and their associated microbial communities form a resilient, interconnected unit, is essential for effective mangrove ecosystem management and conservation (*Allard et al., 2020*; *Morris, 2018*). In this study, we investigate the interaction between microbial communities and hydrological patterns within the mangrove sediments of the southern Gulf of Mexico, focusing on the Pargo estuary in Campeche, Mexico (*Pérez-Ceballos et al., 2022*). We aim to elucidate how hydrological connectivity, the water-mediated exchange of matter, energy, and organisms, and hydroperiod, the frequency and duration of flooding and drying seasons, affect the structure and diversity of these communities. Employing 16S RNA sequencing, we intend to characterize the microbial diversity and structure across different mangrove zones, notably in the fringe, basin, and impaired zones, each exhibiting distinct hydrological characteristics. Here, we considered both the vertical stratification by sampling at various

depths and the temporal dynamics by analyzing data across different seasons. These factors are critical as they are expected to influence microbial assemblages differently. For instance, the fringe zone, regularly washed by tides, likely hosts dynamic microbial populations that can adapt to rapid changes in salinity and water content. In contrast, the basin zone, with less frequent but longer-lasting inundations, and the impaired zone, suffering from disrupted hydrological connectivity, might exhibit microbial communities that have adapted to more static and extreme conditions, such as higher salinity and sulfide concentrations. By integrating depth-specific and seasonal data, we plan to build a comprehensive understanding of how these environmental parameters shape microbial life in mangrove ecosystems. This study not only aims to expand our knowledge of microbial ecology in mangrove sediments but also seeks to provide foundational insights that could inform conservation efforts and the management of these crucial ecosystems under changing global conditions.

## MATERIALS & METHODS

### Study area

Laguna de Términos, designated as a Ramsar site of international and high biological importance, is currently experiencing several pressures due to growing industry and urbanization. The study area encompasses a mangrove forest situated within an estuary known as Estero Pargo on the southeast side of Isla del Carmen in Campeche, Mexico (18°39′05″N; 91°45′31″W and 18°39′03″N; 91°45′26″W, Fig. S1). Estero Pargo, a tidal channel spanning approximately 6 km in length and averaging 14 m in width, covers a surface area of approximately 52 ha. The study area is located approximately 1–2 km away from the nearest urban settlements, with at least two untreated wastewater discharges entering Estero Pargo. Additionally, over the past three years, fish farming activities have expanded within the estuary, along with an increase in the size of a patch of dead trees. Consequently, Estero Pargo represents an urban mangrove that can serve as a model location for studying anthropogenic impacts on biogeochemical cycles in non-pristine mangroves (*Vande Velde et al., 2019*). Portions of this text, as well as additional sections within the methods section, were previously published as part of a preprint (*Vázquez-Rosas-Landa et al., 2024*).

According to the zonation (fringe, basin and impaired area), three mangrove species comprise the vegetation cover in Estero Pargo: at the fringe (approximately 20–30 m wide) along the edge of the tidal channel, the dominant species is the red mangrove (*Rhizophora mangle*), with a few individuals of white mangrove (*Laguncularia racemosa*) also present. Moving to the basin (about 300 m wide), situated behind the fringe, the dominant species shifts to the black mangrove (*Avicennia germinans*). Lastly, there exists a patch (approximately 2.8 ha) of an impaired mangrove area where most of the trees have died, yet a few individuals of *A. germinans* persist (Fig. S1).

The annual average rainfall in the study area is approximately 1,680 mm, with a mean annual temperature of 27 °C (*Coronado-Molina et al., 2012*). The tidal regime is about 0.33 m (*Contreras, Douillet & Zavala-Hidalgo, 2014*). There are two periods of minimum
(February to August) and maximum (September to January) flooding. The soil bulk density is $0.22 \pm 0.008$ g cm$^{-3}$ to $0.44 \pm 0.02$ g cm$^{-3}$ andorganic matter is $8.13 \pm 1.86\%$ to $12.0 \pm 3\%$ (*Pérez-Ceballos et al., 2020*).

## Sampling

To characterize the microbial community in the sediments of Estero Pargo, we conducted sampling on the zones Fringe, Basin and Impared (Fig. S1). We established three sampling plots, each measuring 100 m$^2$, parallel to the tidal channel within each sampling zone. At the midpoint of each plot, we collected sediment cores with a depth of 50 cm from the soil surface. Subsequently, we subsampled each core at intervals of 0–10 cm, 10–30 cm, and 30–50 cm (hereafter referred to as horizon levels) to obtain samples from the central part of each subsection, corresponding to 5 cm, 20 cm, and 40 cm horizon levels. To address seasonal variations, we collected three replicates from each site during two distinct seasons: the flooding season (January) and the dry season (May) in 2018. All samples were transported to the laboratory and stored at $-20\,°C$ until processing (Table S1).

Despite no special permissions were required for soil sampling, we obtained permission to work in the sampling area from the Comisión de Áreas Naturales Protegidas (CONAMP). The permit details are as follows: Dirección de Área Natural Protegida Área de Protección de Flora y Fauna Laguna de Términos, permit No. F00.7.RPCGM.DAPFFLT/441/23.

## DNA extraction

After thawing and fully homogenizing the samples, we extracted DNA for molecular analyses using 250 mg of sediment. To ensure a comprehensive representation of the bacterial community at each horizon level, we pooled five replicates of DNA extraction per core section for analysis. We purified DNA from 54 sediment samples according to the manufacturer's instructions using the E.Z.N.A Soil DNA kit (Omega Biotech, Inc., Norcross, GA, USA).

## Amplification and sequencing

Genetic libraries were prepared using the primers 341F (CCTACGGGNGGCWGCAG) and 785R (GACTACHVGGGTATCTAATCC) to amplify fragments of the V3–V4 regions of the 16S RNA gene (*Ma et al., 2021*). DNA samples were prepared for targeted sequencing with the Quick-16S™ NGS Library Prep Kit (Zymo Research, Irvine, CA, USA). The sequencing library was prepared using real-time PCR machines to control cycles and limit PCR chimera formation. All PCR reactions were set to a final volume of 20 uL and treated with the following thermal protocol: 10 min at 95 °C for denaturation, then 20 cycles of 95 °C for 30 s, 55 °C for 30 s, and 72 °C for 3 min. The final PCR products were cooled at 4 °C prior to quantification with qPCR fluorescence readings and pooled together based on equal molarity. The final pooled library was cleaned up with the Select-a-Size DNA Clean & Concentrator™ (Zymo Research), then quantified with TapeStation® (Agilent Technologies, Santa Clara, CA, USA) and Qubit® (Thermo Fisher Scientific, Waltham, MA, USA). We used the ZymoBIOMICS® Microbial Community Standard (Zymo Research) as a positive control for each DNA extraction. The final library was

sequenced on Illumina® MiSeq™ with a v3 reagent kit (600 cycles), and performed with >10% PhiX spike-in.

## Data processing and taxonomy assignment

The quality of the raw reads was analyzed using FastQC (v.0.12.0) (*Babraham Bioinformatics, 2024*). Subsequently, TrimGalore (v.0.6.10) (*Krueger et al., 2023*) was used to remove Illumina Universal Adapter (AGATCGGAAGAGC) from the reads and Cutadapt to remove primers (*Martin, 2011*). We utilized R (v4.2.2) (*R Core Team, 2024*) and DADA2 (v.1.28) (*Callahan et al., 2016*) to process the reads and calculate the Amplicon Sequence Variants (ASVs). Reads were filtered using the following parameters: truncating the forward reads to 260 bases and the reverse to 200. The maximum expected errors (maxEE) which refers to the limit of errors in a sequence at a base level allowed per read, was set at two errors per read. Reads with an expected error exceeding this value were excluded. The ambiguous bases (N) were not allowed (Table S2). For taxonomy assignment, we employed the function assignTaxonomy from the package DADA2 (v.1.16) (*Callahan et al., 2016*) with the database GreenGenes2 (v.2022.10) (*McDonald et al., 2023*) (Table S3). All the code used for this analysis and the following are located at GitHub (https://github.com/mirnavazquez/Mangroves_01).

## Community analysis

We explored the microbial community accounting for horizon depth, zone and season variables. Visualization of each parameter option was achieved through non-metric multidimensional scaling (NMDS) analysis, employing Bray–Curtis distance. Based on the exploratory analysis, the sample zr2502_16_R1 which could potentially have skewed the analysis, was excluded (Fig. S2). We then computed the prevalence of each feature (ASV) across the samples, incorporating corresponding taxonomy and abundance data. To ensure reliable data, we first filtered out taxa with low prevalence, defined as those present in fewer than two samples, as well as those with unidentified phyla. This step ensured that only ASVs present in a minimum number of samples were considered. Next, we applied a prevalence threshold of 2%, filtering out ASVs whose relative abundance across samples was lower than 2%. This step ensures that only ASVs with sufficient presence across samples are retained. The filtered data were normalized by transforming the ASV table into proportions relative to the total abundance for each sample (Table S4).

## Alpha diversity analysis

We used the estimate_richness function from the phyloseq R package (*McMurdie & Holmes, 2015*) to estimate the alpha diversity of microbial communities across different zones, seasons, and depths. The diversity metrics calculated included Observed, Chao1, Shannon, and Simpson indices. These indices were used to evaluate both the richness and evenness of microbial communities under varying environmental conditions (*i.e.,* different zones, seasons, and depths). The resulting diversity measures were stored in a data frame for further statistical analysis (Table S5).

## Normality assessment

To assess whether the data followed a normal distribution, we performed the Shapiro–Wilk test for each diversity metric (Observed, Chao1, Shannon, and Simpson) within each combination of zone, season, and depth. The shapiro.test function from R base package was applied to each group individually to determine if the distribution of each metric deviated significantly from normality (Table S6).

## Statistical analysis

As some groups did not meet the assumption of normality for certain metrics, we used the Kruskal–Wallis test, a non-parametric method, to compare diversity metrics (Observed, Chao1, Shannon, Simpson) across the different levels of zone, season, and depth. This test was applied to each diversity metric separately, using the kruskal.test function from R base package, to evaluate whether there were significant differences between the groups defined by the factors (Table S7).

## Pairwise comparisons

For pairwise comparisons between the different levels of zone, season, and depth, we applied the Wilcoxon rank-sum test, implemented in the wilcox.test function from the R base package. This test was performed for each pair of groups within each diversity metric (Observed, Chao1, Shannon, Simpson). To control for multiple comparisons, we adjusted the $p$-values using the Bonferroni correction *via* the p.adjust function from the R base package. The results of these pairwise comparisons were summarized and presented in a table for clearer interpretation (Table S8).

## Visualization

To visually present the differences in alpha diversity across the various zones, seasons, and depths, we created boxplots for each diversity metric (Observed, Chao1, Shannon, Simpson) using the ggplot2 package (*Wickham, 2016*). Each plot was annotated with significance levels based on the Wilcoxon rank-sum test results. The significance thresholds were defined as $p \leq 0.001$, $p \leq 0.01$, $p \leq 0.05$, and ns for non-significant results.

## Beta diversity analysis

To assess the factors influencing microbial community structure, we performed a Beta diversity analysis using Bray–Curtis dissimilarity. The analysis was followed by permutational multivariate analysis of variance (PERMANOVA) and principal coordinate analysis (PCoA) to determine the significance of environmental factors such as zone, season, and depth. The sample metadata was extracted from the phyloseq object and converted into a data frame for ease of manipulation. Factors such as zone, season, and depth were ensured to be correctly set as factors within the dataset. We calculated the Bray–Curtis dissimilarity matrix using the phyloseq distance function. This metric is used to quantify the compositional dissimilarity between sample pairs based on their microbial abundance profiles.

## PERMANOVA and interaction effects

We performed a series of PERMANOVA models using the adonis2 function to evaluate the main effects of zone, season, and depth, as well as their interactions. A total of 9,999 permutations were used to assess the significance of each factor (Table S9). The interaction between all three factors (zone, season, depth) was specifically tested, in addition to pairwise comparisons of zone levels using the pairwise.adonis function. To assess the homogeneity of dispersion across different environmental categories, we conducted a Beta dispersion analysis using the betadisper function (Fig. S3). The results were statistically tested using ANOVA to examine whether dispersion varied significantly across zones. We visualized the results of the Beta dispersion analysis using boxplots generated with ggplot2 (*Wickham, 2016*). These plots illustrate the variation in Bray–Curtis dissimilarity across different zones and depth_groups. All statistical tests, including PERMANOVA and beta dispersion analyses, were based on 9,999 permutations to ensure robustness of the results.

## Depth grouping

In order to investigate the role of depth on community structure, we recategorized the depth variable by grouping the 20 cm and 40 cm depth categories into a new factor, depth_group. We performed additional PERMANOVA analyses to test for differences in community structure based on the new depth grouping and its interaction with the other environmental factors.

## Differential abundance analysis

To analyze the microbial community data, we first loaded the pre-processed phyloseq object, which contains both the abundance data (ASV table) and associated sample metadata, using the phyloseq package in R (*McMurdie & Holmes, 2015*). We extracted the count data matrix from the phyloseq object, ensuring that taxa were represented as rows and samples as columns. To account for the presence of zeros in the data, a pseudo-count of 1 was added to all values, which was done to avoid issues during subsequent transformations. The transformed count data was then used to recreate the phyloseq object. This step incorporated the transformed count data, along with the sample metadata, taxonomic information, phylogenetic tree, and reference sequences. We ensured that relevant factors, such as 'zone,' 'season,' and 'depth,' were appropriately treated as categorical variables.

We performed differential abundance analysis using the DESeq2 package (*Wickham, 2016*; *Love et al., 2017*), which allows for the modeling of count data with respect to various factors. The phyloseq object was converted into a DESeq2 dataset, specifying the experimental design to include factors such as zone, season, and depth. The DESeq2 workflow was used to estimate size factors, dispersions, and perform hypothesis testing on the microbial taxa across the different experimental conditions.

## Variance stabilizing transformation

To facilitate the downstream analysis and visualization of the count data, we applied a variance stabilizing transformation (VST) to the DESeq2 object. This transformation helps to stabilize the variance across samples, particularly for low-count taxa. The VST was

applied with the variance Stabilizing Transformation function from the DESeq2 package (*Wickham, 2016*; *Love et al., 2017*).

## Taxa filtering and annotation

We then filtered the transformed data to retain the top 50 taxa with the highest variance across samples. This step helped to focus the analysis on taxa with sufficient variability for differential abundance testing. For the filtered taxa, we extracted their taxonomic information and generated labels that combined the Phylum, Class, and Family levels for more informative representation.

We also created an annotation data frame from the sample metadata, ensuring that the categorical variables, such as 'zone,' 'season,' and 'depth,' were included. These annotations were used to color the heatmap and facilitate visual interpretation of the microbial community patterns across different experimental conditions.

## Heatmap visualization

A heatmap was generated to visualize the relative abundance patterns of the filtered taxa across samples. Clustering was applied to both rows (taxa) and columns (samples) to identify patterns in microbial composition across the experimental variables. Custom annotation colors were used to represent the different levels of the factors 'zone,' 'season,' and 'depth' within the samples. The heatmap was produced using the pheatmap package (*Wickham, 2016*; *Love et al., 2017*; *Kolde, 2010*), with appropriate adjustments to font size and color scaling for optimal visualization.

## Differential expression analysis for contrasts

We defined several contrasts to investigate differential expression across experimental conditions. These contrasts included pairwise comparisons between different zones (Impaired *vs.* Basin, Impaired *vs.* Fringe, Basin *vs.* Fringe), as well as comparisons based on depth (5 cm *vs.* 20 cm, 5 cm *vs.* 40 cm) and season (flood *vs.* dry). Differential abundance analysis was performed for each contrast using the results function from DESeq2 (*Wickham, 2016*; *Love et al., 2017*), and the significance of each comparison was assessed by filtering the results based on a $p$-value threshold of 0.05 and a log2 fold change greater than 1 or less than $-1$.

## Visualization of differentially abundant taxa

To visualize the enriched ASVs from the various contrasts, we first combined the results from all contrasts into a unified data frame. We merged the differential expression results with the corresponding taxonomic labels and aggregated the log2 fold changes by Phylum. A bar plot was then created to visualize the total log2 fold change for each Phylum, grouped by the comparison zone (*e.g.*, "Impaired *vs.* Basin"). Additional visualizations were created for depth-related contrasts (5 cm *vs.* 20 cm, 5 cm *vs.* 40 cm), and the corresponding taxonomic data was filtered and visualized similarly.

## Metabolic inference

PICRUSt2 *Douglas et al. (2020)* was used for the prediction of the functional abundances, where ASVs are placed into a reference tree with reference genomes; hidden-state prediction

approaches are used to infer the genomic content of sampled sequences using several databases such as enzymes (ECs), KEGG orthologues (KOs), cluster of orthologous genes (COGs) and metabolic pathways (MetaCyc). The default cutoff was used for the Nearest Sequenced Taxon Index (NSTI) of 2.0. The predicted metagenomic functions were represented as Kyoto Encyclopedia of Genes and Genomes (KEGG) Orthology (KO) terms. A transformation was applied to the data by adding 1 to each value to prevent zero counts in the subsequent analyses.

## Functional annotation and pathway assignment

Functional data was annotated with biogeochemical cycles (such as Nitrogen, Methane, and Sulfur metabolism) based on the KO terms. This was achieved by joining the KO with associated metabolic cycles from the DiTing database (*Xue et al., 2021*). Only KOs associated with these specific biogeochemical cycles were retained for further analysis. To facilitate downstream analyses, a list of KO descriptions was retrieved from the KEGG database.

## Differential abundance analysis

Differential abundance analysis (DAA) was performed using the DESeq2 method, implemented in the pathway_daa function from the ggpicrust2 package (*Douglas et al., 2020*), to assess differences in KO abundance across experimental factors, including sample zone, depth, and season. For each factor, data was subsetted by the respective group and analyzed separately. Statistical significance was assessed using adjusted $p$-values with a threshold of $p < 0.05$. Significant KOs were filtered, and the results were annotated with associated biogeochemical cycles.

## Heatmap generation

Heatmaps were generated to visualize the differential abundance of significant KEGG pathways. Data were first subsetted to include only the KOs identified as significantly different between groups. The final heatmap matrix was constructed by joining the filtered KO data with the KEGG descriptions and pathways. Rows (KOs) were clustered using hierarchical clustering, and the heatmaps were annotated with biogeochemical cycles, sample zone, depth, and season information to explore the relationships between microbial community functions and environmental factors using pheatmap package (*Wickham, 2016*; *Love et al., 2017*; *Kolde, 2010*).

For zone-based analyses, the samples were grouped by the ecological zone (*e.g.*, "Fringe," "Impaired," and "Basin"). For depth-based analyses, the sample depth groups were considered (*e.g.*, five cm, 40 cm). Similarly, seasonality was considered by grouping samples into dry and flood seasons. The heatmaps were generated using the pheatmap function, which provided hierarchical clustering of both rows and columns, and displayed color gradients to represent the abundance of KOs across samples.

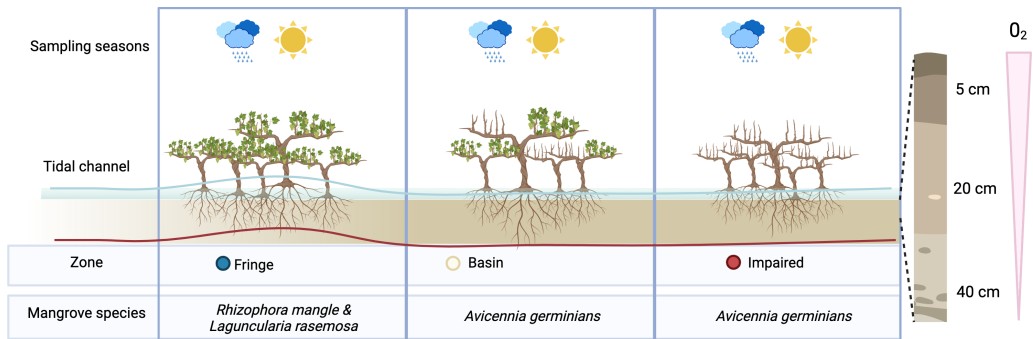

**Figure 1** **Overview of experimental design.** To assess the effect of desiccation on microbial communities, we sampled three distinct zones during two seasons: the flood season (blue line) and the dry season (red line). Microbial communities were examined at three depths: 5 cm, 20 cm, and 40 cm.

## RESULTS

### Microbial diversity dynamics across mangrove ecosystem

We conducted a comprehensive analysis of microbial communities within the mangrove ecosystem using samples collected across different zones, depths, and seasons (Fig. 1). From 665,228 raw sequence reads, we identified 7,987 ASVs, of which 2,561 ASVs were retained after filtering low-abundance taxa.

### Season shape alpha diversity patterns

Season had a significant impact on microbial diversity (Fig. 2), particularly in the Shannon index, with higher diversity observed during the flood season compared to the dry season (pairwise Wilcoxon test $p$-values = 0.012, Figs. 2C, 2E). Zone also influenced microbial diversity, with the Impaired zone exhibiting higher diversity according to the Chao1 index (pairwise Wilcoxon test $p$-values: Impaired $vs.$ Fringe = 0.029, Impaired $vs.$ Basin = 0.023). However, the effect of zone on diversity, while significant, was less prominent than the seasonal effect. Wilcoxon tests revealed that, for the 20 cm horizon, microbial diversity was significantly different between the Impaired and Basin zones in all index except Simpson (Fig. 2E). For the 40 cm horizon, there was no statistically significant difference in Shannon diversity between the Basin and Fringe zones; however, the Impaired zone exhibited a trend toward higher diversity according to the Shannon index compared to both the Basin ($p = 0.078$) and Fringe zones ($p = 0.052$). This suggests that while depth may influence microbial diversity, its effect appears secondary to that of season and zone.

### Zone and depth drive changes in microbial community structure

A Bray–Curtis dissimilarity matrix was computed to assess microbial community dissimilarity, which was used in subsequent multivariate analyses, including PERMANOVA, PCoA, and betadisper analyses. PERMANOVA showed that the interaction of zone, season, and depth significantly influenced microbial community composition ($F = 5.2388$, $p < 0.001$; Table S9). PCoA plots, color-coded by zone, revealed distinct clustering by zone, with significant differences between them (Fig. 3A). The PCoA plots for each zone, faceted

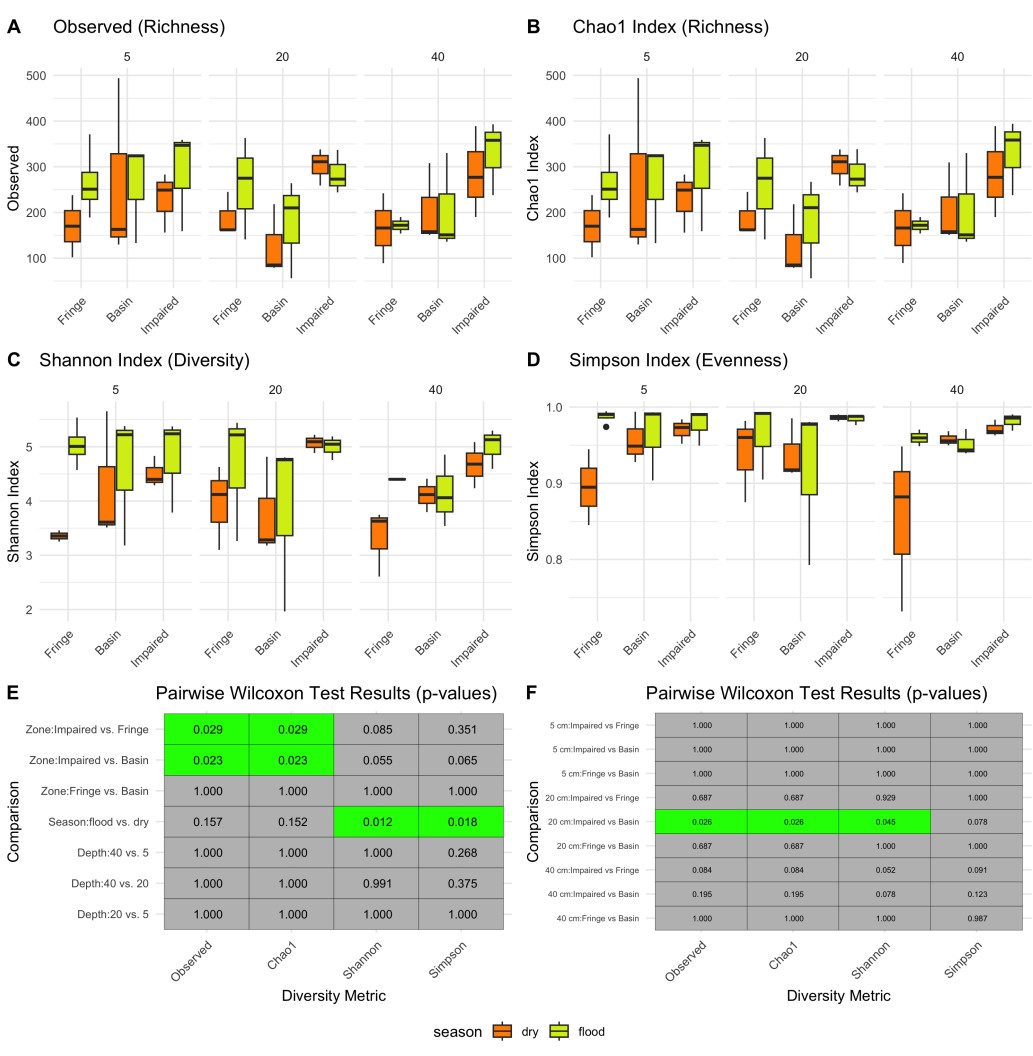

**Figure 2** **Alpha diversity patterns.** Diversity metrics (A–D) and pairwise significance heatmaps (E–F) illustrate the observed richness (A), Chao1 richness (B), Shannon diversity (C), and Simpson evenness (D) across zones, seasons, and depths. Panels (E) and (F) present pairwise Wilcoxon test results for diversity metric comparisons, highlighting significant (green) and non-significant (gray) differences, with $p$-values displayed in each tile.

by season and depth, demonstrated clear separation, indicating significant differences in community structure among the zones (Fringe, Basin, and Impaired) as indicated by the statistical results (Fig. 3C). Betadisper analysis showed significant dispersion differences among zones ($F = 5.9394$, $p = 0.0048$; Fig. S3). Pairwise PERMANOVA confirmed significant differences between all pairs of zones (Fringe *vs.* Basin, Fringe *vs.* Impaired, Basin *vs.* Impaired, all $p < 0.001$). The main effects of zone, season, and depth were significant, with zone explaining the largest proportion of variation ($R^2 = 0.3579$). Further analysis of season and depth revealed significant variation in microbial communities across both factors, with a significant interaction effect ($F = 2.2932$, $p < 0.001$).
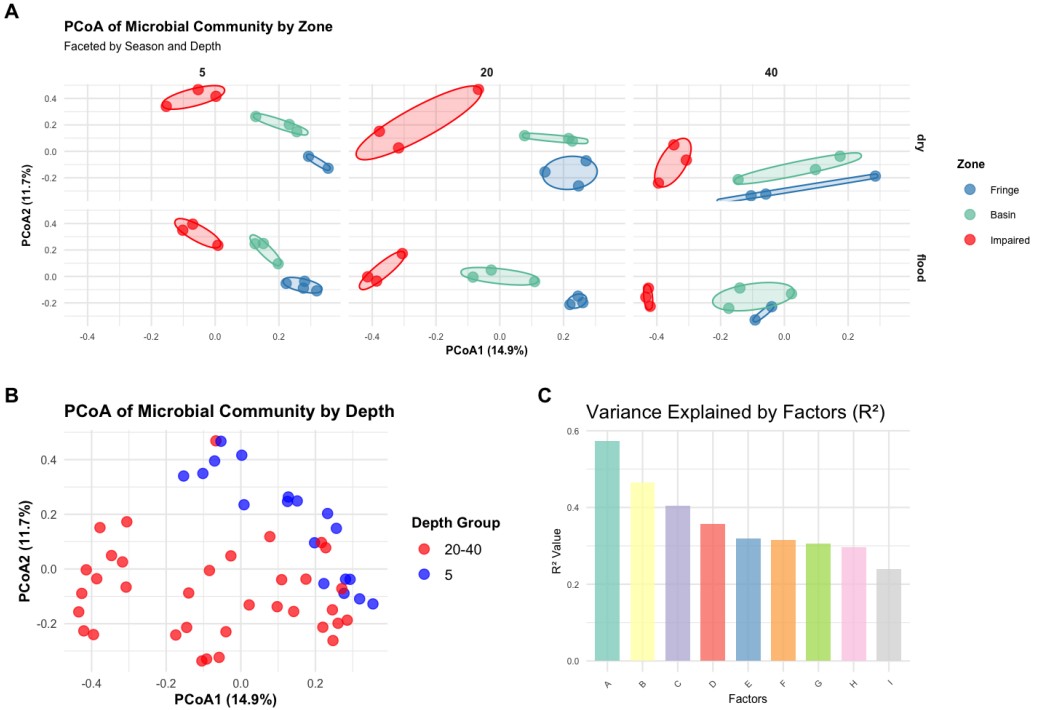

**Figure 3** **Multivariate analysis of microbial community composition and variance explained by environmental factors.** (A) Principal coordinates analysis (PCoA) of Bray–Curtis distances showing microbial community differences across zones, faceted by season and depth. The axes represent the first two principal coordinates, with percentages indicating the variance explained. (B) PCoA of microbial communities grouped by depth, highlighting distinctions in community structure among depth groups. (C) Variance explained ($R^2$ values) by environmental factors, illustrating the relative contributions of individual factors to community composition. Factors labeled as A through I correspond to the following: (A) Interaction of zone, season, and depth, (B) interaction of zone, season, and depth group, (C) interaction of zone and depth, (D) main effect combined of zone, season, and depth, (E) interaction of zone and depth group, (F) main effect combined of zone, season, and depth group, (G) main effect combined of zone and depth, (H) interaction of zone and season, and (I) main effect combined by zone and season.

A new depth grouping variable (depth_group) combining depths of 20 cm and 40 cm into a single category was created. PERMANOVA showed a significant effect of depth group on microbial community composition ($F = 4.0974$, $p = 0.0001$). PCoA plots, color-coded by depth_group, revealed distinct clustering of samples by depth group, with a noticeable separation between the five cm and 20–40 cm depth groups (Fig. 3B), suggesting different microbial community structures between shallow and deeper depths.

## Microbial shifts across zones, depths, and seasons highlight key taxa and ecological patterns

To elucidate microbial diversity and community structure, we performed a differential abundance analysis using DESeq2 and hierarchical clustering of variation. Both analyses revealed distinct patterns across zones, seasons, and depths (Figs. 4 and 5). Across zones, Proteobacteria, particularly Vibrionaceae, were prevalent in Fringe and Basin zones, with a subset restricted to the Fringe. In contrast, Marinomonadaceae and Halomonadaceae
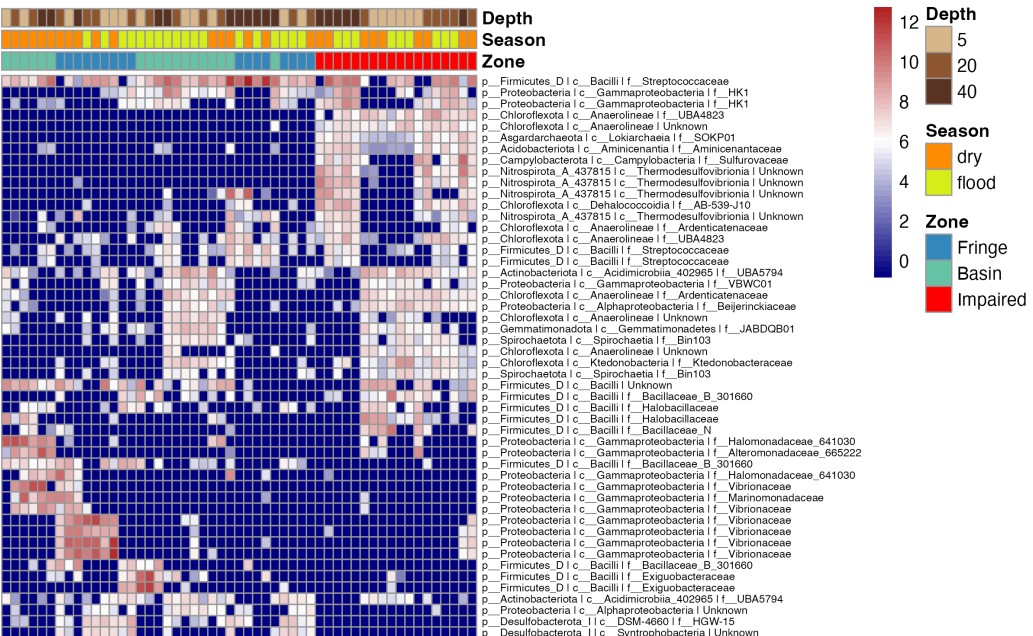

**Figure 4  Heatmap of the top 50 ASVs with the highest variation.** Blue indicates ASVs with low variation, meaning their abundance remains relatively consistent across samples, while red indicates highly variant ASVs, where abundance differs significantly between samples. The heatmap is hierarchically ordered along both the *Y* and *X* axes.

dominated the Basin, while specific lineages, such as Chloroflexota, Asgardarchaeota, Acidobacteriota, Campylobacterota, and Nitrospirota, were exclusive to the Impaired zone. Firmicutes, particularly Streptococcaceae, were uniformly distributed across zones, indicating their ecological importance. Seasonally, Exiguobacteraceae dominated Basin zones during the flood season, while Marinomonadaceae and Halomonadaceae were more abundant in the dry season. Depth-related patterns were particularly pronounced in the Impaired zone. Surface layers (5 and 20 cm) were dominated by Gammaproteobacteria, including Marinomonadaceae, Halomonadaceae, and Vibrionaceae, while deeper layers (20 and 40 cm) harbored lineages such as *Thermodesulfovibrionia*. Streptococcaceae were most prevalent at 40 cm across all zones and seasons.

Differential abundance analysis (Fig. 5) highlighted additional patterns. The Impaired zone exhibited high abundances of *Thermodesulfovibrionia* (Nitrospirota), Lokiarchaeia (Asgardarchaeota), and Sulfurovaceae, while the Fringe zone was enriched in Vibrionaceae and Desulfobacteriota (Syntrophobacteria, Fig. 5A). Depth-related contrasts (Fig. 5B) revealed *Bathyarchaeia*, *Thermodesulfovibrionia*, and *Thermococci* dominating deeper layers, while Gammaproteobacteria and Firmicutes were prevalent at the surface.

## Sulfur and nitrogen metabolism inform us about the metabolic differences among fringe and impaired zones

Phylogenetic placement of ASVs in a genome tree, combined with differential analysis of KO abundances in key pathways (*e.g.*, methane, nitrogen, and sulfur metabolism), revealed

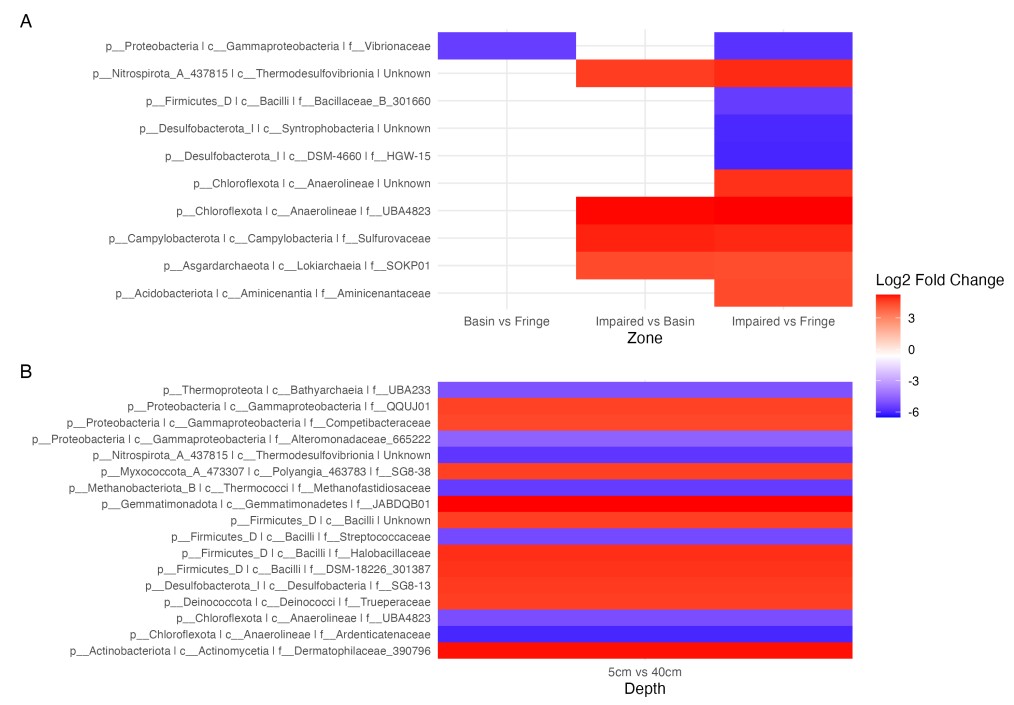

**Figure 5** **Contrast of the top 20 most variable ASVs.** (A) Comparison of ASVs between the experimental condition (fringe and basin) and the control condition (basin and impaired). ASVs present in the experimental condition are shown in blue, and those present in the control condition are shown in red. (B) Comparison of depths, with five cm as the control depth contrasted against the other depths.

contrasting patterns between the Fringe and Impaired zones (Fig. 6). In the Fringe zone, we identified pathways associated with nitrate reduction, while in the Impaired zone, genes related to denitrification were predominant. For sulfur metabolism, sulfate reduction genes were more prominent in the Fringe zone, whereas genes involved in sulfite and DMS oxidation were enriched in the Impaired zone.

In terms of methane metabolism, processes related to methane production from acetate, methylamines, and methanol were more abundant in the Fringe zone. Conversely, methane generation from CO2 and trimethylamines was more common in the Impaired zone. Depth also influenced the distribution of key genes. At the 5 cm horizon, genes related to methane oxidation, such as K10944 (pmoA/amoA; methane/ammonia monooxygenase subunit A (EC:1.14.18.3, 1.14.99.39)) and K10946 (pmoC/amoC; methane/ammonia monooxygenase subunit C), were more prominent. In contrast, genes linked to sulfite oxidation, such as K05301 (sorA; sulfite dehydrogenase subunit A (EC:1.8.2.1)), were more abundant at the 40 cm horizon.

Seasonal variations also influenced metabolic processes. During the dry season, nitrate dissimilatory processes were more prominent compared to the flood season. These processes included genes such as K02568 (napB; nitrate reductase (cytochrome), electron transfer subunit), K00372 (nasC/nasA; assimilatory nitrate reductase catalytic subunit

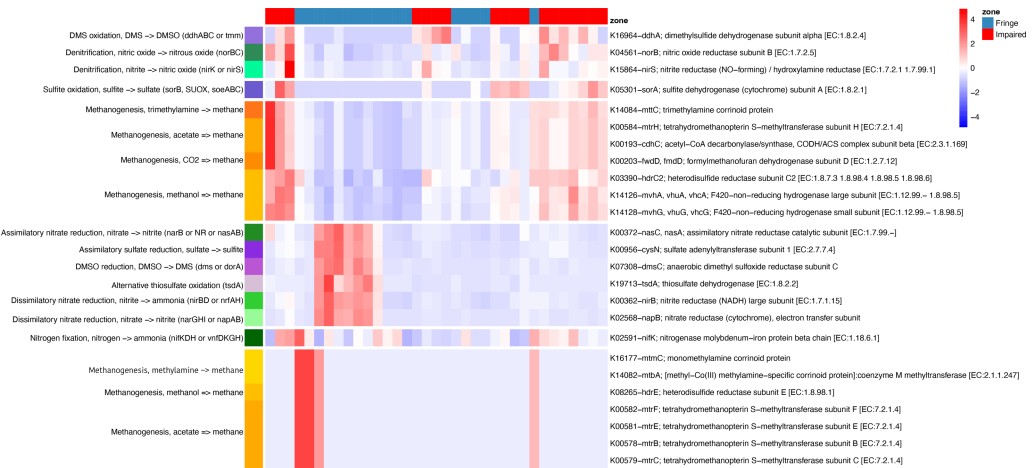

**Figure 6** **Heatmap of KEGG ortholog abundance across fringe and impaired zones.** The heatmap displays the scaled abundance of KEGG Orthologs (KOs) associated with nitrogen, sulfur, and methane metabolism in Fringe and Impaired zones. Rows represent KOs, annotated with their corresponding biogeochemical pathways, while columns represent individual samples grouped by zone. The hierarchical clustering highlights patterns of functional differentiation between the zones. Blue-to-red gradient indicates low to high abundance levels, with significant KOs (adjusted $p$-value $< 0.05$) determined using DESeq2. This visualization reveals the metabolic pathways driving differences in microbial activity across zones.

(EC:1.7.99.–)), K00362 (nirB; nitrite reductase (NADH), large subunit (EC:1.7.1.15)), and K00370 (narG/narZ/nxrA; nitrate reductase/nitrite oxidoreductase, alpha subunit (EC:1.7.5.1, 1.7.99.–)).

# DISCUSSION

Our investigation into the microbial communities of mangrove sediments in the southern Gulf of Mexico underscores the profound impact of hydrological connectivity, or its loss, on microbial diversity and community structure, demonstrating significant variations across different zones, depths, and seasons. This enriches our understanding of microbial dynamics in mangrove ecosystems, particularly in light of established baseline data from this same site (*Pérez-Ceballos et al., 2022*).

## Impact of season and depth on microbial diversity and predicted metabolic pathways

Seasonal changes significantly influence microbial communities, with diversity notably higher during the flood season due to enhanced nutrient availability and reduced salinity (*Pérez-Ceballos et al., 2022*). This environmental condition promotes the growth of diverse taxa, particularly Firmicutes, which thrive under nutrient-rich conditions. Metabolic profiles reveal strong seasonal patterns in genes associated with sulfur cycling, such as *soxB*, *soxZ*, which are found in sulfur-oxidizing microorganisms, and *dsrA* characteristic of sulfur reduction (*Wu et al., 2021*), which are more abundant during the flood season, particularly in the Fringe zone. This suggests shifts toward enhanced sulfur oxidation

and sulfate reduction pathways. Conversely, the dry season promotes anaerobic sulfur metabolism, such as *tsdA* and *dmsC* which is part of the dimethyl sulfoxide reductase system which reduces dimethyl sulfoxide (DMSO) to dimethyl sulfide (DMS) (*Jenner et al., 2019*; *Lubitz & Weiner, 2003*), highlighting the resilience and adaptability of these microbial communities to environmental changes.

Conversely, the dry season fosters oligotrophic conditions that favor halophilic taxa such as *Halomonas* and *Marinomonas*. Alpha diversity peaks during the flood season, especially in the Basin, where Firmicutes dominate. In contrast, Halomonadaceae and Vibrionaceae become more prevalent in the dry season, indicating their adaptability to saline environments. Notably, *Exiguobacterium* shows a strong preference for lower salinity and increased nutrient availability during floods, reflecting its specific ecological niche (*Edbeib, Wahab & Huyop, 2016*; *Chen et al., 2017*).

Consistent with findings from our study, Vibrionaceae have often been linked to areas with high nutrient contamination and urban pollution, suggesting their ability to thrive in eutrophic conditions (*Chen et al., 2020*). In our study, these bacteria were more prevalent in the basin and fringe zones of mangrove ecosystems, areas that typically experience variable hydrological inputs and may be subject to runoff from urban landscapes. This observation is supported by research indicating that Vibrionaceae can dominate microbial communities in nutrient-rich, polluted environments due to their versatile metabolic capabilities and rapid response to organic enrichment. Furthermore, the prevalence of Vibrionaceae in these zones could also reflect their adaptability to the dynamic salinity and nutrient gradients that characterize these mangrove interfaces (*Chen et al., 2020*). This aligns with previous studies that have noted the proliferation of *Vibrio* species in similar settings, underscoring the impact of hydrological and anthropogenic influences on microbial community compositions in coastal ecosystems (*Banchi et al., 2022*).

## Depth-related microbial stratification and ecological implications

Depth plays a crucial role, with clear microbial stratification observed. Surface layers (5 cm) predominantly feature aerobic microbial communities, while deeper layers (20–40 cm) harbor taxa like *Bathyarchaeota*, *Thermococcaceae*, and *Streptococcaceae*, adapted to anaerobic conditions (*Zhou et al., 2018*; *Ranawat & Rawat, 2017*; *Rombouts et al., 2020*). This stratification suggests distinct ecological niches within mangrove sediment profiles, influenced by oxygen availability and nutrient gradients. The presence of specific taxa such as *Thermodesulfovibrionia* (*Umezawa et al., 2021*) in certain zones and depths indicates niche specialization, which could have significant implications for nutrient cycling and ecosystem health. The ecological roles of these taxa, particularly their contributions to nitrogen and sulfur cycles, need further exploration to understand their potential in ecosystem functioning and bioremediation efforts.

## Zone-specific community structures and ecological implications

Microbial community structures vary significantly across the Fringe, Basin, and Impaired zones, reflecting the ecological impacts of environmental conditions. The Impaired zone, characterized by anoxic conditions and enriched in genes related to denitrification and

sulfur oxidation pathways, shows strong microbial adaptation to reduced environments. The presence of *Thermodesulfovibrionia* underscores its role in sulfur cycling under anaerobic conditions. However, this adaptation does not exclude the activity of sulfate reduction, which could still be occurring at a lower yet detectable level. In contrast, the association of the Fringe zone with nitrate reduction pathways suggests a more dynamic and oxygenated environment conducive to aerobic microbial processes. It is crucial to consider that sediment stratification may influence these observations, allowing the coexistence of both anaerobic and aerobic processes in different sediment layers.

Moreover, the results highlight significant interaction effects between zone, season, and depth, underscoring the complexity of microbial community dynamics. These interactions suggest that microbial responses to environmental changes are multifaceted and influenced by a combination of local and temporal factors. Understanding these interactions is crucial for predicting changes in microbial communities in response to ongoing environmental pressures.

Unexpectedly, the Basin site dominated by *Avicennia germinans* displayed a microbial composition distinct from previous descriptions, with lower abundances of *Desulfatiglans* and other lineages typically associated with this mangrove species (*Gómez-Acata et al., 2023*). This discrepancy may be attributed to localized environmental factors, human impacts, or differences in sampling methodologies, emphasizing the unique microbial dynamics at this site.

### Methane metabolism and microbial adaptability

Methane metabolism pathways also vary by depth and site, with the Fringe zone showing methane production from methylamine, methanol, and acetate, while the Impaired zone relies more on $CO_2$ and trimethylamines. Surface layers are enriched with methane oxidation genes (*pmoA/amoA*, *pmoC/amoC*), indicating active methane cycling, whereas deeper layers show higher abundances of sulfite oxidation genes (*sorA*), consistent with increased sulfate-reducing activity at depth. The widespread presence of *Streptococcaceae* across all zones underscores their potential role in biogeochemical processes, affecting methane and carbon cycles (*Doyle et al., 2019*).

### Methodological considerations and future research directions

While we acknowledge the limitations inherent in predicting metabolic functions from 16S rRNA data, the use of PICRUSt2 has provided valuable insights. Future research should incorporate metagenomic and metatranscriptomic techniques to validate these predictions and to explore the full spectrum of microbial functions, particularly in understanding how different taxa contribute to nitrogen, sulfur, and carbon cycling under varied environmental conditions.

## CONCLUSION

This study demonstrates the dynamic nature of microbial communities in mangrove ecosystems, significantly influenced by hydrological connectivity, seasonal water saturation, and sediment depth. By directly contrasting these results with foundational research such as

that conducted in Pargo estuary (*Pérez-Ceballos et al., 2022*), we deepen the understanding of how environmental factors shape microbial landscapes in these critical habitats. These insights are vital for future conservation and management strategies aimed at preserving mangrove ecosystems amidst environmental change and urbanization pressures.

## ACKNOWLEDGEMENTS

We thank Josefina Santos Ramírez, Tomás Zaldívar Jiménez, Ricardo Ortegón Herrera, Mario Alejandro Gómez Ponce, Hernán Álvarez Guillén, and Andrés Reda Deara for their assistance with logistics and field data collection. We also thank Citlalli Garrido Abreu for her contributions to the laboratory analyses. We acknowledge the use of ChatGPT (developed by OpenAI) for assistance with language clarity. The authors remain solely responsible for the content, interpretations, and conclusions presented.

### Funding

The authors received no funding for this work.

### Competing Interests

The authors declare there are no competing interests.

### Author Contributions

- Mirna Vázquez-Rosas-Landa conceived and designed the experiments, analyzed the data, prepared figures and/or tables, authored or reviewed drafts of the article, and approved the final draft.
- Rosela Pérez-Ceballos conceived and designed the experiments, performed the experiments, authored or reviewed drafts of the article, and approved the final draft.
- Arturo Zaldívar-Jiménez conceived and designed the experiments, performed the experiments, authored or reviewed drafts of the article, and approved the final draft.
- Stephanie Hereira analyzed the data, prepared figures and/or tables, authored or reviewed drafts of the article, and approved the final draft.
- Leonardo Pérez González analyzed the data, prepared figures and/or tables, authored or reviewed drafts of the article, and approved the final draft.
- Alejandra Prieto-Davó performed the experiments, authored or reviewed drafts of the article, and approved the final draft.
- Omar Celis-Hernández performed the experiments, authored or reviewed drafts of the article, and approved the final draft.
- Julio Cesar Canales-Delgadillo conceived and designed the experiments, performed the experiments, authored or reviewed drafts of the article, and approved the final draft.

### Data Availability

The amplicon sequences are available at Zenodo: GitJCCD. (2025). GitJCCD/sediment_sequences: row_amplicons_v1.1 (rowseqs1.1). Zenodo. https://doi.org/10.5281/zenodo.14885372.

## Supplemental Information

Supplemental information for this article can be found online at http://dx.doi.org/10.7717/peerj.19371#supplemental-information.

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
