# Peer review of "Impact of seasonal flooding and hydrological connectivity loss on microbial community dynamics in mangrove sediments of the southern Gulf of Mexico"

_PeerJ, doi:10.7717/peerj.19371_

## Round 0.1 · original submission · Major Revisions

Your manuscript has now been reviewed by three external reviewers. The reviewers think that your work is of potential interest; however, they also raised several concerns about your study. In particular, multiple reviewers have concerns about (1) the appropriateness of the title, (2) limitations of functional prediction using PICRUSt2. Apart from these, please also (1) share your codes or adequately describe the parameters used in the bioinformatics analysis for reproducibility, and (2) loosen the 10% filtering criterion used or examine if using such a criterion has any significant effect on your results.

·

Basic reporting

Clear and unambiguous professional English used throughout.
Literature references, sufficient field background/context provided.
Professional article structure, figures, tables. Raw data shared.
The title is a little misleading as it gives higher weight to the flood dynamics of the estuary, while the text focuses more on the degradation levels in relation to prokaryotic diversity loss. Additionaly, the importance of hydrological dynamics is deeply covered in the introduction, but the methods show the study is carried out in two seasons and there is no clear relation between the seasons and the hydrological componet.
Finally, the paper uses a lot of computational analysis but no codes were supplied and the methods section does not adequately describe the parameters used for the analysis.

Experimental design

The manuscript brings original research and fits the scope of the journal.
Research questions are interesting and well defined, although the title should address all the patterns investigated.

Considering the investigation performed, I would require the authors to share their codes, especially because they are using a lot of small packages with highly customizable parameters. It is not clear why these packages were used when single simple packages like Qiime2 or Mothur are available. Why not using one of the Microbial Ecology pipelines which are a good fit for your research?

This leads to further issues like when the authors opt to use ANOVA without previously performing a normality test. For this sort of data, we usually have non-normal distribution, making it important to choose non-parametric analysis such as Kruskal-Wallis test, which is more adequate to compare more than two groups.

Methods are described but with insufficient detail to be replicated and could be better organized. For example, the authors only mention they cluster the sequences into ASVs later on in the methods and that information should be part of the Data Processing section.

Additionally, the filter parameters applied are removing over 94% of the ASVs prior to subsequent diversity analysis, which is a huge loss of information. This filter requires that each ASV be present in at least 10% of all the samples, however since the samples originate from different sites and conditions this has the effect of removing any ASV that is unique to a single site or condition, which seems to be the vast majority of ASVs. If the filter is necessary, I would suggest that the authors first subset the groups by site/condition and then apply the filter to the subgroups, or you may be missing an entire site/condition and will lose resolution when comparing them for differentiation in community structure.

Validity of the findings

I thank the authors for providing the raw data and metadata, however your supplemental files need more descriptive metadata like the taxonomy table showing the ASVs per sample. Since the authors use many packages for their analyses, it is especially important to supply details of the input for each analysis, along with the codes.

It is also important to note that PICRUSt2 has many known limitations, such as limited accuracy outside of the human gut microbiome and housekeeping genes (Sun et al. Microbiome, 2020: 10.1186/s40168-020-00815-y) and limited resolution power (Toole et al. Applied Soil Ecology. 2021: 10.1016/j.apsoil.2021.104129) that make the analysis of metabolic inference less relevant for your results. While there are ways to mitigate false positive identifications, such as only using high quality NSTI assignments, it is not clear if the authors are using any quality control steps. I would ask the authors to reconsider inferring the effects of degradation on metabolic aspects of communities with such an indirect measurement.

Data analysis could be improved in the following ways:

1) Consider substituting the use of many small packages with a pipeline like QIIME2 which is free and covers the majority of steps from data processing to Alpha and Beta diversity. Phyloseq, which is used for some analysis in R is also a very complete package which can be used for most of your visualization. These packages offer robust default options and reduce data format errors from moving inputs and outputs between packages. By using the default parameters you're also less likely to choose the wrong statistical analysis for your data.
2) Share your code and use less packages for your analysis and that will make it easier for quality assessment.
3) Change your filtering parameters as I am sure that throwing away 94% of your ASVs is going to result in artifacts and make your analysis much less precise.

After addressing these issues, the results and discussion are probably going to change enough that I won’t make comments at this point. That said I urge the authors to revise and resubmit as I am enthusiastic to review the revised version.

Additional comments

I like how the authors develop the text going through the context to hypothesis and the analysis proposed, which are in line with the research questions. My comments are more focused on how to improve data analysis in order to obtain better resolution and a high quality paper.

Reviewer 2 ·

Basic reporting

The results are not relevant since it does not adequately describe the sites studied, it mentions that there is a hydrological alteration but does not show any description.

Experimental design

It does not show the study area on a map to locate the sites or the site studied, it is very important because it has been observed that the bacterial community changes in the different ecological types of mangroves, this is due to the characteristic hydroperiod of each one.

It does not describe any variable of the hydroperiod, nor does it describe physicochemical parameters of the sites to be able to see the reason for the differences. It does not make any description of the mangrove community.

Validity of the findings

Although the microbial community changed, but by not relating it to any variable of the hydroperiod, as well as physical and chemical parameters, it cannot reach the conclusions described in the document.

Additional comments

Although the topic is relevant, there is a need to provide an adequate description of the sites with respect to ecological type, description of the hydroperiod, that is, an adequate description of the mangrove studied.

·

Basic reporting

no comment

Experimental design

no comment

Validity of the findings

no comment

Additional comments

Major
1. Vibrionaceae is very common in marine and brackish environments. There are non-pathogenic species as well. Why did you say this species indicates urban contamination (L.46). Figure 5 also showed a high proportion of Vibrionaceae in the non-degraded zone. The discussion in L.505-513 also indicated that Vibrio is high in degraded land, but this was different from your findings.
2. Did you normalize the ASV abundance among different samples in Figure 5? Did you calculate the percent relative abundance for a particular species in all samples? This means the occurrence of each species is compared among different samples, but the occurrence cannot be compared among other species. Please write a clearer explanation.
3. Some species of Vibionaceae were not prevalent in ND. Such exceptional was also found for the Marinomonadaceae and Halomonadaceae. Please explain the characteristics of the distinct samples.
4. L.49 said that no reduction in functional diversity according to the degradation level. However, Figures 6 B and C showed many functional reduction pathways for the degraded samples.
5. Figure 6: I think you should make a COGS plot, like in Fig 6A, for three data types of your study (depth, season, zone). Their EC and PATH graphs can be moved to the supplementary materials. The comparison in Figure 6B and C should be also prepared for the three data types.
6. Figure 6B and C: Can you compare the metabolism abundance for all three location samples in the same figure? Is it necessary to compare only pair samples? If this is the case, the comparison between MD and D should be added.
7. L.421-431: You should check your results in more detail and compare them with other studies.
8. Please explain the limitation of metabolic prediction using 16S rRNA in the discussion. The discussion in L431-434 needs more clarification.
9. The title should be more specific. Flooding is not only the main criterion in your study. Depth and degradation levels were also used. Moreover, you used the season variation factor which is not the flooding pattern. The location of this study should be added too because only one area in one country was studied.

Minor
1. Keywords: hidroperiod  hydroperiod?
2. Is Estero Pargo inside Laguna de Terminos? Please write the connection between them around Lines 140-142.
3. How far between A B and C? Please write the scale in Supplementary Figure 1. Please indicate in the methodology too how far between three replicated places in each location.
4. Supplementary tables should be able to stand alone from the main manuscript. Please explain the sample abbreviation briefly in the table titles.
5. Fig.1: The Overview of Experimental Design is not necessary as you have explained many times in the text. But please add about the dominant plant species in each location in the text too. This information is missing in the text.
6. L.278-279: Are the raw sequence read number (619,326) and ASV number (11,469) combined from all 54 samples? The number was quite low. Or are they the average value from all samples? Please indicate the range of read numbers for all samples.
7. Since the diversity indices are one of the important issues in this study. Please add equations relating to these analyses in the methodology.
8. Fig 2: Please show statistically significant value signs in Figure 2.
9. Fig 5: Please indicate the bacterial phyla in the figure too to see the connection with the main text.
10. L.353-355: Nitrospirota and Lactococcus cannot be seen in Figure 5. Please indicate the place to see this information.
11. Supplementary Table S1: Please use the same terms for the Zone types in Column C. There is a column in Spanish language, and please change it to English. Can you discuss the distinct species among a paired sample using P? Should we use P value or P adjust, please explain.
12. L.373: Figure 4 is not a phylogenetic tree.
13. Please italicize the scientific names in the whole manuscript

---

## Round 0.2 · Minor Revisions

While both reviewers of the previous version of the manuscript agree that the current version is much improved, a few minor concerns still need to be addressed.

·

Basic reporting

The manuscript has been much improved compared to the first submission. The authors have revised the text, including the title, and the text now sounds more cohesive and pleasing for the reader. Overall, the majority of the revisions have been addressed and more importantly, the metadata and codes have been shared. While I feel that the discussion could be a little more developed, this version of the manuscript presents the quality necessary to be published in PeerJ.

Experimental design

The experimental design is now well described.

Validity of the findings

No comment

·

Basic reporting

The author significantly revised the manuscript and made all data accessible online, enhancing the study’s validity, clarity, and quality. Several key issues in microbial ecology were newly interpreted. I appreciate the authors’ efforts in this revision. Below are some comments to further improve the manuscript. Line numbers correspond to the track change file.

Experimental design

no comment

Validity of the findings

1. L.620-622: Supplementary Table 4 does not include PERMANOVA. Please verify.
2. L.626: How can the statistical results be interpreted from Figure 3C?
3. Figure 3C: How are parameters assigned to each factor? How was this plot generated?
4. Figure 5: The title legend explaining control and experimental conditions is unclear. The experimental conditions (fringe and basin) were compared to control conditions (basin and impaired), so control conditions should be denominators and indicated in blue (negative fold change). The current explanation is contradictory—please recheck the analysis.
5. Figure 5B: Why weren’t depth comparisons made between 5 vs. 20 and 20 vs. 40?
6. Sulfate reduction and denitrification typically dominate in anoxic or anaerobic zones. However, the impaired zone shows low sulfate reduction but high denitrification, which appears inconsistent. Please provide comments on this observation.
7. L.973-986: Where can the results of predominant functions at different depths and seasons be found?
8. L.1051-1052: Nitrate reduction occurs under anoxic or anaerobic conditions. Please verify this statement.

Additional comments

9. Abstract: The abstract lacks concrete findings. Key results, including numerical values, microorganism names, or pathway names, should be included.
10. Supplementary Fig. 1 and L.182: Are A, B, and C associated with the fringe, basin, and impaired zones? Please use consistent terminology.
11. Methodology: The section contains too many subtopics. Consider summarizing, merging some sections, or moving detailed information to the supplementary materials.
12. L.895-897: The explanation of major microorganisms inhabited at different layers is incomplete. Either list all names in the chart or indicate that only representative taxa are shown.
13. Figures 5 and 6: The names of microorganisms, mechanisms, and enzymes are too long and small. Consider adjusting the format for readability.
14. Vibrionaceae distribution: Many studies link Vibrionaceae to high nutrient contamination and urban pollution. However, this study found them more prevalent in the basin and fringe zones, aligning with some prior research. It would be interesting to include comments on this observation.
These references may be useful.
https://doi.org/10.1007/s11356-022-22752-z
https://doi.org/10.3389/fmicb.2020.610974
https://doi.org/10.3389/fmicb.2014.00038

---

## Round 0.3 · accepted · Accept

The authors have addressed all of the reviewers' comments. This manuscript is now ready for publication.